# Improving the Performance of Horseradish Peroxidase by Site-Directed Mutagenesis

**DOI:** 10.3390/ijms20040916

**Published:** 2019-02-20

**Authors:** Diana Humer, Oliver Spadiut

**Affiliations:** TU Wien, Institute of Chemical, Environmental and Bioscience Engineering, Research Area Biochemical Engineering, Gumpendorfer Straße 1a, 1060 Vienna, Austria; diana.humer@tuwien.ac.at

**Keywords:** *E. coli*, recombinant horseradish peroxidase, site-directed mutagenesis, periplasm, glycosylation sites

## Abstract

Horseradish peroxidase (HRP) is an intensely studied enzyme with a wide range of commercial applications. Traditionally, HRP is extracted from plant; however, recombinant HRP (rHRP) production is a promising alternative. Here, non-glycosylated rHRP was produced in *Escherichia coli* as a DsbA fusion protein including a Dsb signal sequence for translocation to the periplasm and a His tag for purification. The missing *N*-glycosylation results in reduced catalytic activity and thermal stability, therefore enzyme engineering was used to improve these characteristics. The amino acids at four *N*-glycosylation sites, namely N13, N57, N255 and N268, were mutated by site-directed mutagenesis and combined to double, triple and quadruple enzyme variants. Subsequently, the rHRP fusion proteins were purified by immobilized metal affinity chromatography (IMAC) and biochemically characterized. We found that the quadruple mutant rHRP N13D/N57S/N255D/N268D showed 2-fold higher thermostability and 8-fold increased catalytic activity with 2,2’-azino-*bis*(3-ethylbenzothiazoline-6-sulphonic acid) (ABTS) as reducing substrate when compared to the non-mutated rHRP benchmark enzyme.

## 1. Introduction

The enzyme horseradish peroxidase (EC 1.11.1.7) is a heme-containing oxidoreductase used in both research and diagnostics for a wide range of applications (e.g., immunoassays, diagnostic kits, probe-based assay techniques as ELISA, EMSA, Western blotting and Southern blotting, waste water treatment and as a reagent in organic synthesis [1,2,3,4,5,6,7]. This 308 amino acid metalloenzyme incorporates two calcium atoms and four disulphide bridges [8,9]. In the plant, HRP also contains a hydrophobic 30 amino acid N-terminal leader sequence and a 15 amino acid C-terminal extension [10]. The C-terminal sequence is a sorting signal responsible for secretion to the vacuole [11] and the N-terminal sequence directs the enzyme to the endoplasmatic reticulum (ER) for post translational modifications, namely heme and calcium incorporation, disulphide bond formation and *N*-glycosylation [12]. The asparagine-linked glycans represent about a fifth of the 44 kDa plant holoenzyme. HRP possesses nine potential glycosylation sites and the pattern as well as the occupation of these sites is heterogeneous between HRP variants [13,14]. At least 28 different native HRP isoforms have been described so far [15] out of which HRP C is the most abundant and therefore the most studied one [9]. 

The commercially available HRP is extracted from *Armoracia rusticana* roots. However, only seasonal availability and long cultivation times paired with low yields make the classical production process undesirable. Moreover, the content of single isoenzymes is often very low and downstream processing is tedious. As a consequence, there is a need for a uniform enzyme preparation with defined characteristics and recombinant protein production would mitigate this issue. Many studies have addressed *Saccharomyces cerevisiae* or *Pichia pastoris* as host organisms for rHRP production because yeasts are easy to cultivate and commonly used for glycosylated and disulphide bond containing proteins. Alas, in yeast hyper-glycosylation occurs and the downstream process is cumbersome. Prokaryotes on the other hand lack the organelles necessary for glycosylation, namely ER and Golgi apparatus. In addition, the reducing environment in the cytoplasm of bacteria impedes the formation of disulphide bridges. Hence, recombinant glycoproteins with disulphide bridges are usually not produced in bacteria. However, *E. coli* is a very convenient host organism because of its cheap and easy cultivation at high cell densities. Moreover, there are no obstacles due to hyper-glycosylation as it is the case in yeast. This substantially facilitates downstream processing and allows application of the product for therapeutic use. It has also been shown that glycosylation is not essential for HRP activity or folding [16] although enzyme activity and thermal stability are considerably reduced when compared to the native enzyme [15,17]. Many studies have already been performed with the goal to enhance the general stability and activity properties of rHRP (Table 1). 

Lin et al. [21] identified a N255D mutant by random mutagenesis with 14-fold higher activity than the non-mutated benchmark enzyme but they concluded that this increase was due to better folding of the enzyme rather than improved catalytic performance. Directed evolution was used to identify mutants 13A10 and 17E12 (for mutant descriptions see Table 1, Footnotes) in *P. pastoris,* which were associated with increased specific activity with ABTS (5.4-fold and 2.8-fold) and guaiacol (2.4-fold and 1.2-fold) as substrates. The thermostability of 13A10 was comparable to the non-mutated benchmark enzyme but it was remarkably decreased in 17E12 mutants [27]. Variant 13A10 was used as starting point for successive rounds of directed evolution and gave rise to 13A7, H2-10G5 and 13A7-N175S (for mutant descriptions see Table 1, Footnotes). These variants were found to be more stable towards pH, temperature, SDS, urea and sodium chloride but enzyme activity was not further improved [26]. Ryan et al. [23] intensely studied the influence of site-directed mutations on hydrogen peroxide tolerance. They identified T110V, K232N and K241F, which were 25-, 18- and 12-fold more resistant towards hydrogen peroxide than the non-mutated benchmark enzyme. These variants also showed increased tolerance to heat and solvents. In addition, K232N as well as K241F displayed higher turnover numbers (k_cat_) with ABTS as reducing substrate [22]. Asad et al. [18] changed the amino acids present at two *N*-glycosylation sites of rHRP by site-directed mutagenesis. They described variants N13D and N268D, which showed increased catalytic efficiency with phenol/4-aminoantipyrine and were both more stable in terms of hydrogen peroxide and heat tolerance. A follow-up study identified N268G, which showed 18-fold higher resistance towards hydrogen peroxide and 2.5-fold higher thermal stability [20]. Capone et al. [19] performed a profound investigation of *N*-glycosylation mutants in *P. pastoris*, where asparagines at eight sites were replaced by aspartic acid, serine or glutamine. They showed that the positive influence of N13D and N268D on thermal stability is also valid for expression in yeast and that the variant identified by Lin et al. [21] was apparently slightly beneficial in terms of catalytic activity. Interestingly, a variant with mutations at all eight *N*-glycosylation sites showed substantially decreased activity and thermal stability.

At the moment, HRP is not used for in vivo medical applications, because the plant glycosylation pattern differs significantly from human glycoforms and therefore has immunogenic potential [28]. This can be circumvented by reducing the glycosylation pattern to mannose-type glycans, which can be achieved by adding an ER retention sequence. Unfortunately, these glycans lead to rapid clearance from circulation in humans [28]. Nevertheless, a combination of HRP and paracetamol or the plant hormone indole-3-acetic acid (IAA) was found to be medically active in targeted cancer treatment [29,30]. Although this cytotoxic effect has been known since the nineties [31,32], up to now, HRP is not considered suitable for therapeutic use. For this application, a recombinantly produced single isoform free of glycosylation with sufficient stability and activity would be desirable. In this study, we investigated the *N*-glycosylation mutants N13D, N57S, N255D, N268D and combinations thereof for thermal stability and catalytic efficiency with the substrate ABTS for rHRP expressed in *E. coli.* Soluble rHRP was preferred for mutant screenings because refolding of rHRP from *E. coli* inclusion bodies is a complex and cumbersome procedure which still has to be optimized. Therefore, we chose an expression system that leads to translocation of rHRP into the periplasm. The aim of this work was to improve the traits of non-glycosylated rHRP towards higher stability and catalytic efficiency to increase suitability for medical applications. Indeed, a promising rHRP *N*-glycosylation mutant was identified and biochemically characterized.

## 2. Results and Discussion

### 2.1. Protein Production

Recombinant HRP variants were produced and then translocated to the periplasm by the DsbA signal sequence. Soluble proteins were isolated from the periplasm and rHRP was purified to gain active, correctly folded enzyme. The imidazole concentration in the IMAC binding buffer was at the upper limit given by the column manufacturer (GE Healthcare, Chicago, IL, USA) to avoid unspecific interactions between *E. coli* host cell proteins and the stationary IMAC phase. Nevertheless, several impurities were visible on the SDS PAGE of the IMAC eluate (data not shown). Therefore, rHRP concentrations were calculated using an SDS-PAGE HRP standard curve with known concentrations (Appendix A). The peak area of rHRP was determined using Fiji Image Analysis Software (https://fiji.sc) [33] and the protein content was calculated using the slope of the linear regression line of known rHRP concentrations. This led to final enzyme titres of 0.05–0.09 mg rHRP g^−1^ DCW. The final rHRP product yield was between 0.04–0.08 g L^−1^ and is similar to reported values from Gundinger et al. [17] for soluble rHRP in pET39b^+^ (0.048 g L^−1^). 

### 2.2. Biochemical Characterization

#### 2.2.1. Biochemical Characterization of Benchmark rHRP and Seven rHRP Mutants

##### 2.2.1.1. Enzyme Kinetics

Plant HRP VI-A (Sigma-Aldrich, St. Louis, MO, USA), non-mutated benchmark rHRP and the seven rHRP variants N13D, N57S, N255D, N268D, N57S/N268D, N57S/N255D/N268D and N13D/N57S/N255D/N268D were analysed for steady-state kinetics with ABTS as reducing substrate. The kinetic constants are presented in Table 2. N13D and N255D showed less catalytic efficiency (K_cat_/K_m_) than the benchmark enzyme and for N13D this is in accordance with Capone et al. [19]. For N255D on the other hand, Capone et al. [19] observed almost the same catalytic activity compared to the benchmark enzyme (1.1-fold increase). N268D had a 2-fold increased turnover number (k_cat_) when compared to the non-mutated rHRP and the same trend was shown by Asad et al. [20], where a 2.6-fold enhanced k_cat_ with phenol/4-aminoantipyrine was reported. The slightly enhanced catalytic efficiency of N57S, when compared to the benchmark rHRP (1.2-fold), is in accordance with Capone et al. [19] (1.4-fold). The triple mutant N57S/N255D/N268D reached a 3.2-fold higher turnover number than the benchmark rHRP and with a 10-fold increase, N13D/N57S/N255D/N268D showed the highest fold change of k_cat_ when compared to the benchmark rHRP. However, only N57S and N13D/N57S/N255D/N268D had an increased catalytic efficiency when compared to the non-mutated rHRP (1.2-fold and 2-fold, respectively). In general, the results were greatly affected by the unusually high Michaelis Menten constants (K_m_) and the considerable standard deviations (see Section 2.2.2).

##### 2.2.1.2. Thermal Stability

The thermal stability of plant HRP VI-A, non-mutated benchmark rHRP and the seven rHRP variants was determined at 60 °C (Table 3). N13D and N268D were found to enhance stability towards heat, which is in accordance with Asad et al. [18] and Capone et al. [19]. However, Capone et al. [19] also reported a positive effect of N57S, which could not be confirmed. Variant N255D seemed to have no effect on thermal stability. The double mutant N57S/N268D was similar to the rHRP benchmark enzyme concerning temperature susceptibility, so apparently the effect of the mutations is not additive. N268D had the highest benefit and was 3.6-fold more stable than the non-mutated rHRP, whereas the quadruple mutant N13D/N57S/N255D/N268D was 2.4-fold enhanced. Although N268D was the most thermostable variant, the quadruple mutant was chosen for further investigations, as this variant showed promising results for both catalytic activity and thermal stability.

#### 2.2.2. Catalytic Activity of Plant HRP under Different Conditions

Interestingly, all obtained Michaelis Menten constants (Table 2) were much higher than previously reported for soluble rHRP variants [17] and this was also the case for commercially available plant HRP [17,34,35]. The substantial variability of the obtained Michaelis Menten constants may be a direct result of the high K_m_ values we observed due to fast reactions and high slopes, as variability increases with reaction velocity (see Table 2). We assumed that the measurement buffer was influencing the results of the kinetic measurements, as this was the only apparent difference to the previous procedure from Gundinger et al. [17]. Therefore, the catalytic activity of plant HRP was determined in both buffers: 50 mM BisTris/HCl pH 7, 7% glycerol, 100 mM NaCl and 50 mM KH_2_PO_4_ pH 6 (Table 4). The results indicated that the high Michaelis Menten constants, as well as the high standard deviations, are indeed buffer dependent. The K_m_ value was 6-fold higher in 50 mM BisTris/HCl pH 7, 7% glycerol, 100 mM NaCl buffer when compared to 50 mM KH_2_PO_4_ pH 6. As a consequence, the catalytic efficiency was 9-fold enhanced when the measurements were performed in potassium phosphate buffer. The kinetic parameters observed with 50 mM KH_2_PO_4_ pH 6 are in accordance with Gundinger et al. [17], as they reported a K_m_-value of 1.75 mM and a V_max_-value of 567 U mg^−1^.

However, at this point it was still unclear whether these results were obtained because of the buffer substance, the pH change or the additives in the BisTris buffer. Therefore, the specific activity of plant HRP with ABTS as reducing substrate was examined under different conditions (Figure 1). Apparently, the additives glycerol and sodium chloride have a negative influence, as the specific activity was 2-fold enhanced when they were omitted. The two buffers 50 mM KH_2_PO_4_ pH 7 and 50 mM BisTris/HCl pH 7 led to comparable results. The catalytic activity at pH 5 was 3-fold enhanced when compared to pH 7 when the measurements were conducted in potassium phosphate buffer. Based on these results, we decided to use 50 mM KH_2_PO_4_ pH 5 for all consecutive measurements (Section 2.2.3). 

#### 2.2.3. Optimized Biochemical Characterization of Benchmark rHRP and Mutant N13D/N57S/N255D/N268D

The first investigation (Section 2.2.1) showed that the quadruple mutant N13D/N57S/N255D/N268D was the most promising variant when compared to the non-mutated benchmark rHRP (Table 2 and Table 3). Therefore, a second protein purification and biochemical characterization with optimized assay conditions was performed to confirm these results.

##### 2.2.3.1. Enzyme Activity

Enzyme kinetic measurements with plant HRP, non-mutated benchmark rHRP and rHRP variant N13D/N57S/N255D/N268D were performed in 50 mM KH_2_PO_4_ pH 5 using 96-well plates. Here, N13D/N57S/N255D/N268D showed an 8-fold enhanced catalytic efficiency (K_cat_/K_m_) and an 8-fold increased turnover number (k_cat_) when compared to the benchmark enzyme (Table 5). The increase in k_cat_ for N13D/N57S/N255D/N268D was similar to the results obtained with the previous assay (Table 2, 10-fold). Besides, the K_m_ (0.27 mM) for plant HRP was similar to values reported in literature: 0.27 mM [35] and 0.11 mM [34]. However, the K_m_-values observed for plant HRP in potassium phosphate buffer differed significantly between the two assays (Table 4 and Table 5). We assume that this might be due to differences in pH, as the Michaelis Menten constant in 50 mM KH_2_PO_4_ pH 6 was 1.5 mM (Table 4) and Gundinger et al. [17] reported 1.75 mM in 50 mM KH_2_PO_4_ pH 6.5. Gilfoyle et al. [35] and Grigorenko et al. [34] used sodium phosphate/citrate buffer at pH 5 and sodium acetate buffer at pH 5, respectively, which resulted in K_m_-values of 0.27 mM [35] and 0.11 mM [34] which are in accordance with 0.27 mM in potassium phosphate (Table 5).

##### 2.2.3.2. Thermal Stability

The thermal stability of plant HRP VI-A, non-mutated benchmark rHRP and rHRP variant N13D/N57S/N255D/N268D was determined at 60 °C (Table 6). The measurement was performed at pH 7 to guarantee comparability with the previous assay (Section 2.2.1.2). Interestingly, the half-life of all HRP species measured in 50 mM KH_2_PO_4_ buffer pH 7 was slightly higher than the half-life in 50 mM BisTris/HCl pH 7, 7% glycerol, 100 mM NaCl (Table 4). Asad et al. [36] and Haifeng et al. [37] found that phosphate buffers can influence the thermostability of horseradish peroxidase. Although the samples were kept in 50 mM BisTris/HCl pH 7, 7% glycerol, 100 mM NaCl during heat exposition in both experiments, it might still be possible that potassium phosphate had a stabilizing effect during the activity measurement. In both experiments the plant HRP was far more stable than the rHRP variants (Table 3 and Table 6) because glycans improve enzyme stability [18]. The thermostability of the quadruple mutant N13D/N57S/N255D/N268D was 2-fold higher than the stability of the benchmark rHRP enzyme (Table 6) which is in accordance with the previous assay (Table 3; 2.4-fold).

Summarizing, the quadruple mutant showed a significantly augmented performance concerning thermal stability as well as catalytic activity with ABTS as substrate. Therefore, this variant is considered a good starting point for further enzyme engineering approaches.

## 3. Materials and Methods

### 3.1. Chemicals

Chemicals were purchased from Carl Roth (Karlsruhe, Germany) or AppliChem (Darmstadt, Germany). Plant HRP Type VI-A (Cat. No.: P6782) was purchased from Sigma-Aldrich (St. Louis, MO, USA), enzymes were purchased from New England Biolabs (Ipswich, MA, USA) or Thermo Fisher Scientific Inc. (Waltham, MA, USA), 2,2’-azino-*bis*(3-ethylbenzothiazoline-6-sulphonic acid) (ABTS) was purchased from Sigma-Aldrich or AMRESCO^®^ biochemical (Solon, OH, USA).

### 3.2. Strains and Plasmids

The *hrp* gene coding for HRP variant C1A was codon-optimized for *E. coli* and obtained from GenSript USA Inc. (Piscataway, NJ, USA). HRP was produced as a His6-tagged recombinant protein from pET39b^+^ in the *E. coli* strain BL21 (DE3) (Lucigen, Middleton, WI, USA). The plasmid pET39b+ (Novagen, San Diego, CA, USA) encodes a Dsb tag for export and periplasmic folding, so that a DsbA fusion protein with a HIS tag between the *dsbA* sequence and the *hrp* sequence is generated.

### 3.3. Strain Generation by Site-Directed Mutagenesis

The following plasmids were constructed with standard molecular cloning techniques [38]. Whole plasmid PCR was used to introduce mutations in the *hrp* gene by site-directed mutagenesis. The 7 kb fragment was amplified with the respective oligonucleotides to generate single, double, triple and quadruple mutations (Table 7). All oligonucleotides were purchased from Microsynth (Balgach, Switzerland). Each PCR reaction contained 1× Q5 Reaction Buffer, 200 µM dNTP Mix, 200 nM of both forward and reverse primer, 100 ng template vector DNA and 1 U Q5 High-Fidelity DNA Polymerase. The PCR products were purified with the Monarch PCR & DNA Cleanup Kit from New England Biolabs (NEB, Ipswich, MA, USA) and the template plasmid DNA was removed by FastDigest *Dpn*I (Thermo Scientific™, Waltham, MA, USA) digestion. 1 FDU (FastDigest unit, see Abbreviations) of *Dpn*I was added to the cleaned PCR products and incubated overnight at 37 °C. After heat inactivation at 80 °C for 20 min, the plasmids were transformed into BL21 (DE3). All DNA inserts of the recombinant plasmids were verified by DNA sequencing (Microsynth, Balgach, Switzerland).

### 3.4. Growth Conditions and Protein Production

LB medium (10 g L^−1^ tryptone, 10 g L^−1^ NaCl and 5 g L^−1^ yeast extract) or SB medium (32 g L^−1^ tryptone; 20 g L^−1^ yeast extract; 5 g L^−1^ NaCl; 5 mM NaOH) was used for cultivation of BL21 (DE3) strains. Kanamycin was added to a final concentration of 50 mg L^−1^ to ensure plasmid maintenance. Pre-cultures were grown overnight at 37 °C with shaking (250 rpm) in 50 mL LB^Kan^ or SB^Kan^ medium and 2.5 L Ultra Yield Flasks (UYF) were inoculated to reach an optical density (OD_600_) of 0.3 in a final volume of 500 mL LB^Kan^ or SB^Kan^ medium. The cells were grown at 37 °C with shaking (250 rpm) until an OD_600_ of 0.5, subsequently *hrp* expression was induced by adding 0.1 mM isopropyl β-d-1-thiogalactopyranoside (IPTG). After growth for 20 h at 25 °C and 250 rpm, the cells were harvested by centrifugation (4500 rpm, 30 min, 4 °C). 

### 3.5. Protein Purification 

The cell pellets were resuspended in buffer A (50 mM BisTris/HCl pH 7, 500 mM NaCl, 40 mM imidazole) with cOmplete™ Protease Inhibitor Cocktail (Roche, Basel, Switzerland). The cell suspension was homogenized with an Avestin Emulsiflex C3 high pressure homogenizer (Avestin, Ottawa, ON, Canada) for 10 passages at 1000 bar and centrifuged afterwards at 10,000 rpm for 1 h. Particles were removed from the supernatant by filtration (0.2 µm) prior to protein purification with IMAC using the Äkta pure system (GE Healthcare, Chicago, IL, USA). The column (HisTrap^™^ Fast Flow Crude 1 mL, GE Healthcare) was equilibrated with 10 column volumes (CV) buffer A and the crude extract was loaded with a linear flow rate of 156 cm h^−1^. Subsequently, the column was washed with 10–20 CV buffer A before step elution with 100% buffer B (50 mM BisTris/HCl pH 7, 500 mM NaCl, 500 mM imidazole) at a linear flow rate of 156 cm h^−1^. The eluted protein fractions were desalted with Sephadex G-25 PD-10 desalting columns (GE Healthcare, Chicago, IL, USA) and eluted in buffer C (50 mM BisTris/HCl pH 7, 7% glycerol, 100 mM NaCl). Glycerol and sodium chloride were added to the buffer as these substances were found to positively impact on rHRP stability (data not shown). Total protein content was measured using the Bradford assay [39] and the eluate was analysed by SDS PAGE. The concentration of rHRP was calculated as µM mL^−1^ using 61.5 kDa (with DsbA Protein and His tag) as molecular mass of the fusion protein. Hemin and calcium chloride were added in a 2-fold and 4-fold molar amount, respectively and the enzyme preparations were incubated over night at 4 °C with slight agitation. 

### 3.6. SDS PAGE

SDS PAGE was performed according to the Laemmli protocol [40]. Mini-PROTEAN^®^ TGX Stain-Free^™^ Precast Gels (Bio-Rad, Hercules, CA, USA) were used and the gel was run with SDS running buffer (25 mM Tris, 200 mM glycine, 0.1% SDS) in a Bio-Rad Mini-PROTEAN^®^ Tetra Cell. Proteins were separated at 125 V for 1 h and the bands were visualized with Coomassie Brilliant Blue solution. Bio-Rad Precision Plus Protein™ Dual Xtra Prestained Protein Standard or Thermo Scientific PageRuler™ Plus Prestained Protein Ladder, 10 to 250 kDa (Waltham, MA, USA) were used as mass standards. The SDS PAGE was analysed using a ChemiDoc™ MP System with Image Lab™ Software (Bio-Rad, Hercules, CA, USA). 

### 3.7. Biochemical Enzyme Characterization 

#### 3.7.1. Assay Development

##### 3.7.1.1. Pathlength Correction for Microplates

The pathlength for 200 µL reaction volume was determined experimentally by measuring the absorbance of MilliQ water in the Tecan Infinite M200 PRO (Tecan, Männedorf, Switzerland) and the Hitachi U-2900 spectrophotometer (Hitachi, Tokyo, Japan). Water absorption can be measured at near infrared wavelength with a maximum absorbance at 975 nm, the measurement at 900 nm subtracts background absorbance (e.g., plastic of 96-well plate). The correct pathlength for the 96-well plates was then calculated according to Equation (1):
(1)$$pathlength=\frac{A975\;\mathrm{nm}\;\left(well\right)-A900\;\mathrm{nm}\;\left(well\right)}{A975\;\mathrm{nm}\;\left(cuvette\right)-A900\;\mathrm{nm}\;\left(cuvette\right)}\times 10\;\mathrm{mm}$$
Thus, a reaction volume of 200 µL results in a pathlength of 0.58 cm in 96-well plates. 

##### 3.7.1.2. Determination of Extinction Coefficient for ABTS

The extinction coefficient of the ABTS radical was determined experimentally with a Tecan Infinite M200 PRO system (Tecan, Männedorf, Switzerland) for ABTS purchased from Sigma-Aldrich (St. Louis, MO, USA) and AMRESCO^®^ biochemical (Solon, OH, USA). ABTS concentrations ranging from 0 mM to 0.15 mM were measured in 50 mM KH_2_PO_4_ pH 5 buffer with 1 mM hydrogen peroxide in a final volume of 200 µL and 22.5 ng plant HRP Type VI-A (Sigma-Aldrich, Cat. No.: P6782). The reaction was followed to the plateau phase in a 96-well plate at 420 nm for 40 min at 30 °C. Finally, the extinction coefficient was calculated from the slope of the linear regression line of ABTS concentration plotted against maximal absorbance. The extinction coefficient of ABTS at 420 nm is 27 mM^−1^ cm^−1^ for ABTS purchased from both companies (Sigma-Aldrich: y = 15.455x + 0.0548; R^2^ = 0.9977/ AMRESCO^®^ biochemical: y = 15.455x + 0.0314; R^2^ = 0.9986). 

##### 3.7.1.3. Enzyme Activity Calculation

HRP activity in Units mL^−1^ (1 Unit is defined as the amount of enzyme which oxidizes 1 µmol of ABTS per minute) was calculated according to Equation (2):
(2)$$\frac{U}{\mathrm{mL}}=\frac{\left(\Delta A420\;\mathrm{nm}-\Delta A420\;\mathrm{nm}\;blank\right)\times total\;volume\;of\;reaction\times d}{\varepsilon \times p\times \mathrm{sample}\;\mathrm{volume}}$$
*d* = dilution factorp = pathlength in cm, which is 1 for cuvettes and 0.58 in a 96-well plate when the reaction volume is 200 µL (see Section 3.7.1.1).ε = 27 mM^−1^ cm^−1^


#### 3.7.2. Catalytic Activity of Plant HRP under Different Conditions

Enzyme activity of plant HRP Type VI-A (Sigma-Aldrich, Cat. No.: P6782, St. Louis, MO, USA) was determined either in 50 mM BisTris/HCl pH 7, 7% glycerol, 100 mM NaCl or in 50 mM KH_2_PO_4_ pH 6. The reaction mixture in the cuvette contained 27.6 ng HRP, a saturating hydrogen peroxide concentration of 1 mM and varying ABTS concentrations (0.1–10 mM) in 50 mM BisTris/HCl pH 7, 7% glycerol, 100 mM NaCl or in 50 mM KH_2_PO_4_ pH 6 with a final volume of l mL. The increase in absorption was followed at 420 nm for 180 s at 30 °C in a Hitachi U-2900 spectrophotometer (Hitachi, Tokyo, Japan). All measurements were performed in triplicates. Kinetic parameters were calculated using OriginPro software (OriginLab Corporation 2016, Northampton, MA, USA). As these measurements revealed substantial differences concerning the kinetic parameters between the two buffers, it was decided to further investigate the influence of buffer substance, pH and buffer additives in a 96-well plate format. Here, the reaction mixture in each well contained a saturating hydrogen peroxide concentration of 1 mM and 5 mM ABTS in a final volume of 200 µL. Commercially available plant HRP (9 ng; Sigma-Aldrich, Cat. No.: P6782, St. Louis, MO, USA) was added to the reaction mixture and the increase in absorption was followed at 420 nm for 180 s at 30 °C in a Tecan Infinite M200 PRO instrument (Tecan, Männedorf, Switzerland). The following buffers were used: 50 mM KH_2_PO_4_ pH 5; 50 mM KH_2_PO_4_ pH 6.5; 50 mM KH_2_PO_4_ pH 7; 50 mM BisTris/HCl pH 7, 7% glycerol, 100 mM NaCl or 50 mM BisTris/HCl pH 7. All measurements were performed in duplicates.

#### 3.7.3. Enzyme Kinetics

Enzyme kinetic parameters for the substrate ABTS were determined with non-mutated rHRP benchmark enzyme, seven rHRP mutants and commercially available plant HRP Type VI-A. The samples were either measured in cuvettes with a U-2900 spectrophotometer (Hitachi, Tokyo, Japan) or in 96-well plates using a Tecan Infinite M200 PRO instrument (Tecan, Männedorf, Switzerland).

##### 3.7.3.1. Hitachi U-2900 Spectrophotometer 

The reaction mixture in the cuvette contained 20 µL protein sample, a saturating hydrogen peroxide concentration of 1 mM and varying ABTS concentrations (0.1–10 mM) in 50 mM BisTris/HCl pH 7, 7% glycerol, 100 mM NaCl buffer with a final volume of l mL. The increase in absorption was followed at 420 nm for 180 s at 30 °C in a Hitachi U-2900 spectrophotometer. All measurements were performed in triplicates. The kinetic parameters were calculated using OriginPro software (OriginLab Corporation 2016, Northampton, MA, USA).

##### 3.7.3.2. Tecan Infinite M200 PRO 

The reaction mixture in each well of the 96-well plate contained a saturating hydrogen peroxide concentration of 1 mM and varying ABTS concentrations (0.1–10 mM) in 50 mM KH_2_PO_4_ pH 5 buffer in a final volume of 200 µL. Protein sample (5 µL) was added to each well and was filled up with 195 µL reaction mixture. The increase in absorption was followed in a 96-well plate at 420 nm for 180 s at 30 °C in a Tecan Infinite M200 PRO instrument. Eight replicates were used for all measurements. The kinetic parameters were calculated using OriginPro software (OriginLab Corporation 2016, Northampton, MA, USA). 

#### 3.7.4. Thermal Stability

The thermal stability of the enzyme was assessed at 60 °C in 50 mM BisTris/HCl pH 7, 7% glycerol, 100 mM NaCl and the residual activity with ABTS was measured after 0, 1, 2, 3 and 5 min for the non-mutated rHRP benchmark enzyme and the seven rHRP mutants. The residual activity of plant HRP Type VI-A (Sigma-Aldrich, Cat. No.: P6782, St. Louis, MO, USA) was measured after 0, 30, 60, 90 and 120 min, respectively. The protein concentration was normalized to 0.05 g L^−1^ for all samples to minimize potential influences of protein concentrations on thermal stability. The reaction mixture contained 20 µL of protein, a saturating hydrogen peroxide concentration of 1 mM and 10 mM ABTS in 50 mM BisTris/HCl pH 7, 7% glycerol, 100 mM NaCl or in 50 mM KH_2_PO_4_ pH 7 with a final volume of l mL. The increase in absorption was followed at 420 nm for 180 s at 30 °C in a Hitachi U-2900 spectrophotometer (Hitachi, Tokyo, Japan). The residual enzyme activity was plotted against incubation time and the half-life at 60 °C was calculated using the rate of inactivation in Equation (3):
$$t_{1/2}=\frac{ln2}{k_{in}}$$
*t*_1/2_ = half life*k_in_* = slope of the logarithmic residual activity


## 4. Conclusions

HRP has many features that make it suitable for therapeutic use: it is stable at 37 °C, shows high activity at physiological pH and can be conjugated to antibodies or lectins [9]. Thus, it is highly interesting to engineer rHRP for increased activity and stability for use in medical applications. In our study we discovered a new non-glycosylated rHRP variant with improved characteristics by site-directed mutagenesis of amino acids at the *N*-glycosylation sites. N13D/N57S/N255D/N268D was found to substantially increase activity with ABTS as substrate, the catalytic efficiency of this variant was 8-fold higher when compared to the rHRP benchmark enzyme. Moreover, N13D/N57S/N255D/N268D is 2-fold more stable towards high temperature exposure than the non-mutated rHRP. Currently, we work on further improvement of N13D/N57S/N255D/N268D by directed evolution, as well as the additional introduction of selected mutants (see Table 1). Finally, our future goal will be to produce the resulting non-glycosylated rHRP variant in a large-scale inclusion body process.

## Figures and Tables

**Figure 1 ijms-20-00916-f001:**
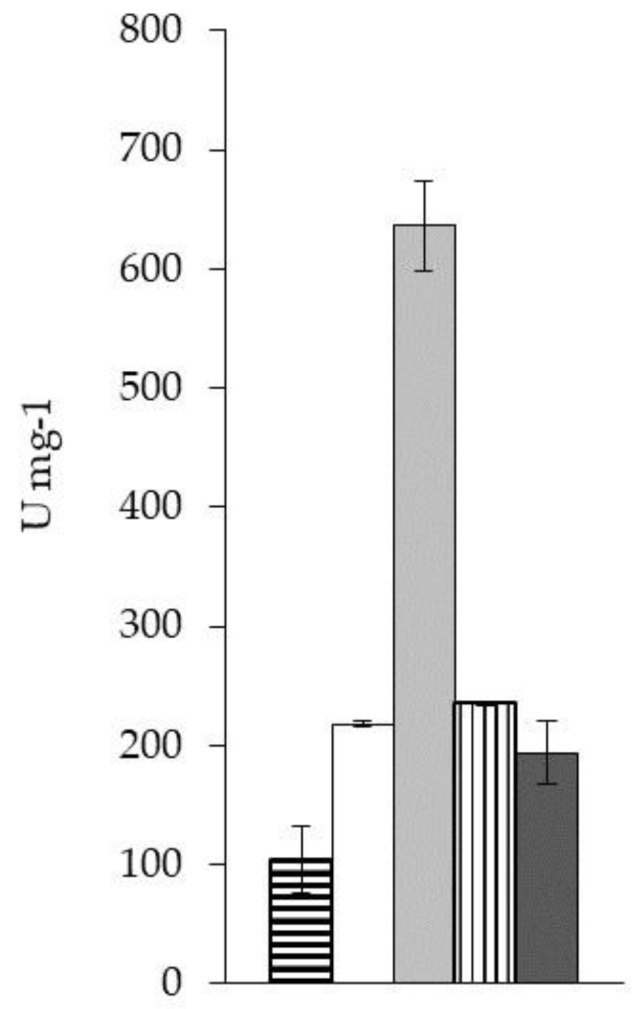
Specific activity of plant HRP Type VI-A with ABTS under various conditions. Five different buffers were used to measure the catalytic activity of plant HRP with 5 mM ABTS and 1 mM hydrogen peroxide. Horizontal stripes: 50 mM BisTris/HCl pH 7, 7% glycerol, 100 mM NaCl; white: 50 mM BisTris/HCl pH 7; light grey: 50 mM KH_2_PO_4_ pH 5; vertical stripes: 50 mM KH_2_PO_4_ pH 6.5; dark grey: 50 mM KH_2_PO_4_ pH 7.

**Table 1 ijms-20-00916-t001:** List of rHRP mutations that improve enzyme performance, listed by authors.

Mutation	Effect	Reference
N13D	Increased stability towards H_2_O_2_ Increased thermal stability	Asad et al. [18] Capone et al. [19]
N268D	Increased stability towards H_2_O_2_ Increased thermal stability Increased substrate specificity for phenol/4-aminoantipyrine Increased activity with phenol/4-aminoantipyrine	Asad et al. [18] Asad et al. [20] Capone et al. [19]
N268G	Increased stability towards H_2_O_2_ Increased thermal stability Increased substrate specificity for phenol/4-aminoantipyrine	Asad et al. [20]
N57S	Increased activity with ABTS Increased activity with H_2_O_2_ Increased thermal stability	Capone et al. [19]
N186D	Increased activity with ABTS	Capone et al. [19]
N198D	Increased substrate specificity for ABTS	Capone et al. [19]
N255D	Better folding in *E. coli* Increased activity with ABTS Increased activity with H_2_O_2_	Lin et al. [21] Capone et al. [19]
N158D	Increased activity with H_2_O_2_	Capone et al. [19]
K232N	Increased activity with ABTS Increased thermal stability Increased solvent stability Increased stability towards H_2_O_2_	Ryan et al. [22] Ryan et al. [23]
K232F	Increased activity with ABTS Increased thermal stability Increased solvent stability	Ryan et al. [22]
E238Q	Increased substrate specificity for ABTS	Ryan et al. [22]
K241N	Increased activity with ABTS	Ryan et al. [22]
K241E	Increased substrate specificity for ABTS Increased activity with ABTS	Ryan et al. [22]
K241A	Increased activity with ABTS Increased stability towards H_2_O_2_	Ryan et al. [22] Ryan et al. [23]
K232N/K241N	Increased thermal stability Increased stability towards H_2_O_2_	Ryan et al. [22] Ryan et al. [23]
K232F/K241N	Increased activity with ABTS Increased thermal stability Increased solvent stability Increased stability towards H_2_O_2_	Ryan et al. [22] Ryan et al. [23]
K232Q/K241Q	Increased activity with ABTS	Ryan et al. [22]
T110V	Increased stability towards H_2_O_2_ Increased thermal stability	Ryan et al. [24] Ryan et al. [23]
T102A	Increased activity with ABTS	Ryan et al. [24]
K232E	Increased stability towards H_2_O_2_	Ryan et al. [23]
K241F	Increased stability towards H_2_O_2_	Ryan et al. [23]
R118K/R159K/K232N/K241F	Increased thermal stability Increased stability towards H_2_O_2_	Ryan et al. [25]
13A7 *	Increased activity with guaiacol	Morawski et al. [26]
H2-10G5 *	Increased activity with guaiacol Increased activity with ABTS Increased pH stability Increased thermal stability Increased stability towards SDS/urea/sodium chloride	Morawski et al. [26]
13A7-N175S *	Increased activity with guaiacol Increased activity with ABTS Increased pH stability Increased thermal stability Increased stability towards SDS/urea/sodium chloride	Morawski et al. [26]
13A10 *	Increased activity with guaiacol Increased activity with ABTS	Morawski et al. [27]
17E12 *	Increased activity with guaiacol Increased activity with ABTS	Morawski et al. [27]

* 13A7 = A85(GCC→GCT)/N212D/Q223L; * H2-10G5 = A85(GCC→GCT)/N175S/N212D; * 13A7-N175S = A85(GCC→GCT)/N212D/Q223L/N175S; * 13A10 = R93L/T102A/L131P/N135(AAC→AAT)/L223Q/T257(ACT→ACA)/V303E; * 17E12 = N47S/T102A/G121(GGT→GGC)/L131P/N135(AAC→AAT)/L223Q/P226Q/T257(ACT→ACA)/P289(CCT→CCA).

**Table 2 ijms-20-00916-t002:** Kinetic characteristics of plant HRP, rHRP and seven rHRP variants with ABTS as reducing substrate measured in 50 mM BisTris/HCl pH 7, 7% glycerol, 100 mM NaCl.

HRP variant	K_m_ [mM]	V_max_ [mol^−1^ L^−1^ × s]	K_cat_ [s^−1^]	K_cat_/K_m_ [mM^−1^ s^−1^]
Benchmark rHRP	2.82 ± 1.52	1.7 × 10^−6^ ± 4.3 × 10^−7^	1.52 ± 0.38	0.54 ± 0.32
N13D	3.29 ± 0.33	8.8 × 10^−7^ ± 4.0 × 10^−8^	1.04 ± 0.05	0.32 ± 0.04
N57S	3.22 ± 0.59	2.7 × 10^−6^ ± 2.6 × 10^−7^	2.10 ± 0.19	0.64 ± 0.13
N255D	4.37 ± 0.86	8.9 × 10^−7^ ± 1.0 × 10^−7^	1.10 ± 0.12	0.24 ± 0.05
N268D	7.85 ± 4.98	2.4 × 10^−6^ ± 1.0 × 10^−6^	3.00 ± 1.25	0.38 ± 0.29
N57S/N268D	4.18 ± 3.55	1.0 × 10^−6^ ± 3.9 × 10^−7^	1.50 ± 0.58	0.36 ± 0.34
N57S/N255D/N268D	9.52 ± 6.89	4.0 × 10^−6^ ± 2.0 × 10^−6^	4.81 ± 2.37	0.51 ± 0.44
N13D/N57S/N255D/N268D	13.6 ± 6.63	1.7 × 10^−5^ ± 5.6 × 10^−6^	15.1 ± 5.00	1.11 ± 0.65
HRP Type VI−A	9.46 ± 5.18	5.7 × 10^−3^ ± 1.8 × 10^−3^	271 ± 87.4	28.7 ± 18.3

**Table 3 ijms-20-00916-t003:** Half-life of plant HRP, rHRP and seven rHRP variants at 60 °C in 50 mM BisTris/HCl pH 7, 7% glycerol, 100 mM NaCl.

HRP Variant	*t*_1/2_ at 60 °C
Benchmark rHRP	2 min 39 s ± 16 s
N13D	3 min 53 s ± 16 s
N57S	2 min 46 s ± 14 s
N255D	2 min 48 s ± 9 s
N268D	9 min 32 s ± 2 min 18 s
N57S/N268D	2 min 14 s ± 56 s
N57S/N255D/N268D	5 min 51 s ± 18 s
N13D/N57S/N255D/N268D	6 min 19 s ± 12 s
HRP Type VI-A	117 min ± 9 min 55 s

***t*_1/2_** = half life.

**Table 4 ijms-20-00916-t004:** Comparison of kinetic characteristics measured with plant HRP Type VI-A in two different buffers.

Buffer	K_m_ [mM]	V_max_ [mol^−1^ L^−1^ × s]	K_cat_ [s^−1^]	K_cat_/K_m_ [mM^−1^ s^−1^]
Buffer 1	9.46 ± 5.18	5.7 × 10^-3^ ± 1.8×10^-3^	272 ± 87.4	28.7 ± 18.3
Buffer 2	1.51 ± 0.15	7.9 × 10^-3^ ± 6.3 × 10^-4^	378 ± 30.0	251 ± 32.2

Buffer 1, 50 mM BisTris/HCl pH 7, 7% glycerol, 100 mM NaCl; Buffer 2, 50 mM KH_2_PO_4_ pH 6.

**Table 5 ijms-20-00916-t005:** Kinetic characteristics of plant HRP, rHRP and N13D/N57S/N255D/N268D with ABTS as reducing substrate in 50 mM KH_2_PO_4_ pH 5.

HRP variant	K_m_ [mM]	V_max_ [mol^−1^ L^−1^ × s]	K_cat_ [s^−1^]	K_cat_/K_m_ [mM^−1^ s^−1^]
Benchmark rHRP	0.44 ± 0.10	2.0 × 10^−6^ ± 9.8 × 10^−8^	2.24 ± 0.11	5.07 ± 1.16
N13D/N57S/N255D/N268D	0.45 ± 0.12	1.8 × 10^−5^ ± 1.1 × 10^−6^	17.4 ± 1.01	39.1 ± 10.5
HRP Type VI-A	0.27 ± 0.05	8.8 × 10^−3^ ± 6.0 × 10^−4^	422 ± 28.9	1,572 ± 306

**Table 6 ijms-20-00916-t006:** Half-life of plant HRP, rHRP and N13D/N57S/N255D/N268D at 60 °C in 50 mM KH_2_PO_4_ pH 7.

HRP Variant	*t*_1/2_ at 60 °C
Benchmark rHRP	3 min 29 s ± 1 s
N13D/N57S/N255D/N268D	7 min 41 s ± 31 s
HRP Type VI-A	133 min ± 1 min 20 s

**Table 7 ijms-20-00916-t007:** Oligonucleotide primers to mutate four Asn residues that act as *N*-glycosylation sites to either Asp or Ser.

N-site	Name	Sequence (5′→3′ Direction)
Benchmark rHRP	pET39b^+^_hrp_fwd	GCGAATGCCCATGGATATGCAACTG
Benchmark rHRP	pET39b^+^_hrp_rev	CCCGGGACTCGAGTTACGAGTT
N13	N13D_fwd2	CTGCCCGGATGTGAGCAACA
N13	N13D_rev2	CGGGCAGCTATTATCATAGAAGG
N57	N57S fwd	CTGCTGGACAGCACCACGTCC
N57	N57S rev	GTCCAGCAGGATACTTGCATCACAGCC
N255	N255D_fwd2	TTAGTTCCCCGGATGC
N255	N255D_rev2	CGGGGAACTAAACAGTTCT
N268	N268D fwd	GTTCGTTCATTTGCCGATTCGACCCAGA
N268	N268D rev	GGCAAATGAACGAACCAGCGGAATCG

The mutated sites are underlined. fwd: forward; rev: reverse.

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
