# Peer review of "Improving the Performance of Horseradish Peroxidase by Site-Directed Mutagenesis"

_ijms, 2019, doi:10.3390/ijms20040916_

Round 1

Reviewer 1 Report

This is a short ms which expands the characterization of mutants of horseradish peroxidase expresed heterologously in E. coli. Numerous studies have proposed and tested different mutants; in this ms, we see the characterization of 7 mutants and the authors select a quadruple mutant for further characterization. Improved activity (8-fold) and thermostability (2.2-fold) are the final results for that quadruple mutant. Due to the previous publication of numerous mutants and attempts to increase activity and thermostability of horseradish peroxidase, the reader would expect some higher improvements that those reported from the results of the study. Besides that comment, the ms described the experiments to characterize the mutants and compare them among them and with results from the literature. A few minor points for correction below.

Lines 44-45. Please remove "is great" by something like: There is a need for a uniform enzyme preparation with defined characteristics and recombinant protein production would mitigate this issue.

Please, do not write the name sof all the authors for most references reported throughout the text of the ms. Use the numerals (indicative for each reference) or at most the first author (and et al.).

Variability of Km results could be a consequence of extremely high Km values obtained with very fast reactions and high slopes).

About the relevance of the study, the fact that plant peroxydase presents much higher stability than rHRP and its mutants reduced greatly the interest on enzymes that have over 20-fold higher activity. Besides, there are numerous studies reporting the characterization of different HRP mutants which rests originality to this study. This ms contributes to know the response of HRP mutant with some additional mutations (4 mutantion in total and two mor ethan reports from a previous study) introduced in the gene. 

Author Response

Reviewer 1

1.                  Lines 44-45. Please remove "is great" by something like: There is a need for a uniform enzyme preparation with defined characteristics and recombinant protein production would mitigate this issue.

2.                  Please, do not write the name sof all the authors for most references reported throughout the text of the ms. Use the numerals (indicative for each reference) or at most the first author (and et al.).

3.                  Variability of Km results could be a consequence of extremely high Km values obtained with very fast reactions and high slopes).

4.                  About the relevance of the study, the fact that plant peroxydase presents much higher stability than rHRP and its mutants reduced greatly the interest on enzymes that have over 20-fold higher activity. Besides, there are numerous studies reporting the characterization of different HRP mutants which rests originality to this study. This ms contributes to know the response of HRP mutant with some additional mutations (4 mutantion in total and two mor ethan reports from a previous study) introduced in the gene.

Points 1 and 2 have been changed in the article (Lines 44 – 46, removal of author names throughout the text). We agree to point 3 where the reviewer states that the variability of the obtained Km results may be a cause of high Km values obtained with fast reactions and high slopes, therefore we added this into the manuscript at line 146.

Point 4: Indeed, for general purposes rHRP is by no means comparable to the plant enzyme, however, this is not our goal, as we aim to produce a non-glycosylated single-isoenzyme HRP preparation which can be used for therapeutic purposes. Under these premises, the result of our initial screening which led to a unglycosylated rHRP variant with 8-fold increased catalytic efficiency is of significance.

Reviewer 2 Report

1. Page 4, line 100-102. Therefore, the rHRP content was calculated from total protein content of the eluate in relation to the rHRP-DsbA band observed on the SDS PAGE gel using Fiji Image Analysis Software [28].

Does this means that the content was calculated after estimating the purity judging from the SDS-PAGE? If so, what is the purity? However, this is not entirely correct since proteins are stained differently on the gel, depending on the composition, so an impurity could seem a huge band but in reality is not that much or a significant impurity not well detected.

A better way of calculation of concentration of HRP would be from the molecular spectra in visible domain, due to the heme cofactor. Establishing the correct concentration is very important since some catalytic parameters (kcat) depend on it.

2. page 4, line 116-124

The authors compare the Vmax, but this depends on enzyme concentration. There is no mention of enzyme concentration. A better way could be to compare kcat.

3. Table 2 and through the manuscript. How could Vmax (maximum reaction rate from MM curve) be measure in U/mg. This unit is rather for specific activity, since it is a rate it has to be something like mol/s or related. The authors have to clarify this. 

4. pag 5, line 152. The exact buffer from the reference could be easily tested, why the authors haven’t done it? The km (could be judged as affinity for the substrate), is rarely depended on commons buffer but the Vmax can depend a lot. So, the authors have to be careful with this.

5. pag 6, line 169-170. It is not surprising, it is known that peroxidases and some oxidase have higher activity as the pH goes down. But, phosphate is not a good buffer at pH 5. A citrate/acetate could be better.

6. How could the authors explain the eq 1. Why this wavelength?

Author Response

Reviewer 2

1. Page 4, line 100-102. Therefore, the rHRP content was calculated from total protein content of the eluate in relation to the rHRP-DsbA band observed on the SDS PAGE gel using Fiji Image Analysis Software [28].

Does this means that the content was calculated after estimating the purity judging from the SDS-PAGE? If so, what is the purity? However, this is not entirely correct since proteins are stained differently on the gel, depending on the composition, so an impurity could seem a huge band but in reality is not that much or a significant impurity not well detected.

A better way of calculation of concentration of HRP would be from the molecular spectra in visible domain, due to the heme cofactor. Establishing the correct concentration is very important since some catalytic parameters (kcat) depend on it.

 2. page 4, line 116-124

The authors compare the Vmax, but this depends on enzyme concentration. There is no mention of enzyme concentration. A better way could be to compare kcat.

 3. Table 2 and through the manuscript. How could Vmax (maximum reaction rate from MM curve) be measure in U/mg. This unit is rather for specific activity, since it is a rate it has to be something like mol/s or related. The authors have to clarify this. 

4. pag 5, line 152. The exact buffer from the reference could be easily tested, why the authors haven’t done it? The km (could be judged as affinity for the substrate), is rarely depended on commons buffer but the Vmax can depend a lot. So, the authors have to be careful with this.

 5. pag 6, line 169-170. It is not surprising, it is known that peroxidases and some oxidase have higher activity as the pH goes down. But, phosphate is not a good buffer at pH 5. A citrate/acetate could be better.

 6. How could the authors explain the eq 1. Why this wavelength?

Point 1: The wording concerning the method for calculating the amount of rHRP in the eluate might have been too ambiguous. We determined all peak areas of a respective SDS-PAGE lane with Fiji Image Analysis Software and calculated the rHRP concentrations from the regression line of a rHRP standard curve with known concentrations where the peak areas were also determined with Fiji Image Analysis Software. The rHRP standard curve is attached in the supplementary data file. We also tried to determine the concentrations with molecular spectra but due to the fact that hemin was added after IMAC purification in a 2-fold molar ratio to the eluate the measured concentrations were not within a reasonable range although we centrifuged prior to the measurements to remove unbound or aggregated hemin.

Point 2: We changed this throughout the manuscript and only compare kcat and kcat/Km.

Point 3: It makes no difference if V [mol/l*s], U/ml or U/mg is used for the Michaelis Menten Plot as it leads to the same Km, kcat and kcat/Km values. Nevertheless, we changed Vmax to mol/L*s throughout the manuscript.

Point 4: We admit that this is true, unfortunately it was not obvious from the beginning that the high variabilities of the obtained values were buffer dependent. However, it was more an issue of the buffer additives than the buffer substance per se.

Point 5: We agree that KH2PO4 buffer is not the first choice, however, we wanted to use this buffer because of comparability with previous studies. At present we are using phosphate-citrate buffer pH5.

Point 6: We explained this more detailed in 3.7.1.1.

Reviewer 3 Report

The authors Humer and Spadiut report on the kinetic and stability parameters of mutant variants of horseradish peroxidase produced in E. coli. Unfortunately, they did not reach any significant increase of activity of the mutants compared to the native plant enzyme (Kcat/Km 4,3 vs. 289, respectively), even though minor increase of activity and thermal stability was observed in the quadruple mutant. Moreover, I have found some mainly experimental issues that should be dealt with before the paper is considered for publication:

In the Introduction the citations on expression of HRP in yeasts and plants are completely missing, please add and discuss some of them.

All over the manuscript, the citation in the text should not comprise the full author list, just the first author et al.

The enzyme is produced as a fusion protein comprising a his-tag and a DsbA-tag, this quite substantially increases the mass of the protein produced and it can also influence the activity of the enzyme. The plasmid used contains a thrombin cleavage site, so that the fusion tags can be cleaved off the prepared HRP. This should be done and the activity of the enzyme void of tags should be tested.

The kinetic data presented in Table 2 are non-sense, as the authors admit that the assays were performed in an unsuitable buffer. So it is clear that all the data must be acquired in the proper buffer system (phosphate buffer pH 5), which gives results relevant to literature reports. The whole table may then look absolutely different. This is the same for the thermostability data shown in Table 3.

Author Response

Reviewer 3

1.                  The authors Humer and Spadiut report on the kinetic and stability parameters of mutant variants of horseradish peroxidase produced in E. coli. Unfortunately, they did not reach any significant increase of activity of the mutants compared to the native plant enzyme (Kcat/Km 4,3 vs. 289, respectively), even though minor increase of activity and thermal stability was observed in the quadruple mutant. Moreover, I have found some mainly experimental issues that should be dealt with before the paper is considered for publication:

2.                  In the Introduction the citations on expression of HRP in yeasts and plants are completely missing, please add and discuss some of them.

3.                  All over the manuscript, the citation in the text should not comprise the full author list, just the first author et al.

4.                  The enzyme is produced as a fusion protein comprising a his-tag and a DsbA-tag, this quite substantially increases the mass of the protein produced and it can also influence the activity of the enzyme. The plasmid used contains a thrombin cleavage site, so that the fusion tags can be cleaved off the prepared HRP. This should be done and the activity of the enzyme void of tags should be tested.

5.                  The kinetic data presented in Table 2 are non-sense, as the authors admit that the assays were performed in an unsuitable buffer. So it is clear that all the data must be acquired in the proper buffer system (phosphate buffer pH 5), which gives results relevant to literature reports. The whole table may then look absolutely different. This is the same for the thermostability data shown in Table 3.

Point 1: As we are mainly interested in using HRP for therapeutic issues, our focus lies on the unglycosylated rHRP and variants thereof. However, some of the studies presented in Table 1 have been performed in yeast (Capone et al., 2014; Morawski et al., 2000; Morawski et al., 2001).

Point 2: This has been acknowledged and changed throughout the manuscript.

Point 3: The study at hand was intended as an initial screening of rHPR mutants with the goal to reduce overall hydrophobicity by mutating the amino acids at the N-glycosylation sites of the enzyme which in theory might lead to enhanced enzyme stability as well as activity (Capone et al., 2014). DsbA catalyzes disulphide bond formation and it has been shown that co-expression of Dsb proteins increases the production of soluble HRP in E. coli (Kondo et al., 2000). DsbA and the HIS tag are influencing all rHRP variants to the same extent, therefore we decided it would be unnecessary to remove the tags during the initial screening.

Point 4: This is true, but we performed a pre-test where we compared the activities of all HRP variants in U/ml with 5 mM ABTS, 1 mM H2O2 in 50 mM BisTris/HCl pH 7, 7% glycerol, 100 mM NaCl with those of 5 mM ABTS, 1 mM H2O2 in 50 mM KH2PO4 pH 5. It can be seen that the trend is the same when the values are compared in relation to the non-mutated rHRP (green = better than the benchmark enzyme, red = worse than the benchmark enzyme) for all variants except HRP N57S/N255D/N268D. Moreover, the trends seen in the phosphate buffer measurements are the same as those for the catalytic efficiency tested in the BisTris buffer: all variants show decreased efficiency except N57S and HRP N13D/N57S/N255D/N268D. As HRP N13D/N57S/N255D/N268D showed by far the best improvement we decided to continue solely with this variant.

BisTris buffer pH7   [U/ml]

KH2PO4 buffer pH5   [U/ml]

Benchmark HRP

0,060

0,22

HRP N13D

0,032

0,03

HRP N57S

0,102

0,26

HRP N255D

0,025

0,01

HRP N268D

0,056

0,10

HRP N57S/N268D

0,034

0,07

HRP N57S/N255D/N268D

0,083

0,17

HRP N13D/N57S/N255D/N268D

0,286

1,50

However, for the thermostability the buffer interferes with the ABTS reaction in general and the thermostability data are obtained by comparing the initial enzyme activity with residual activity after thermal stress. It is a relative loss in activity which uses the initial value as 100%. The loss of activity in % over time does not depend on the extent of catalytic efficiency. BisTris with glycerol and sodium chloride in general is enhancing the stability of the enzyme rather than reducing it (Asad et al., 2013; Eggenreich et al., 2016).

We sincerely hope that this comments have been beneficial and for a positive feedback from your side.

Round 2

Reviewer 2 Report

The authors answered adequately all raised questions and I therefor recommend the manuscript for publication.

For future, I would still recommend the authors to determine the pathlength of a certain volume in the 96-well to use the visible domain (with a known high extinction coefficient compound) since the spectrophotometer has better primary source and detectors for this domain than near IR.

Author Response

We thank the Reviewer for the suggestions!

Reviewer 3 Report

The authors have improved the manuscript at least at some points, the others were justified in a satisfactory way. The paper may be accepted for publication in its current form.

Author Response

We corrected Langugae and Grammar.